# Falling through the cracks: Increased vulnerability and limited social assistance for TB patients and their households during COVID-19 in Cape Town, South Africa

Lieve Vanleeuw[1,2]*, Wanga Zembe-Mkabile[1,3]°, Salla Atkins[2,4,5]°

1 Health Systems Research Unit, South African Medical Research Council, Tygerberg, South Africa,
2 Global Health and Development, Faculty of Social Sciences, Tampere University, Tampere, Finland,
3 Archie Mafeje Social Policy Research Institute, School of Transdisciplinary Research and Graduate Studies, University of South Africa, Pretoria, South Africa, 4 Department of Global Public Health, Karolinska Institutet, Stockholm, Sweden, 5 WHO Collaborating Centre on Tuberculosis and Social Medicine, Karolinska Institutet, Stockholm, Sweden

° These authors contributed equally to this work.
* lieve.vanleeuw@mrc.ac.za

**Data Availability Statement:** Data underlying the results presented in this article cannot be shared

## Abstract

Amid the COVID-19 crisis, Tuberculosis (TB) patients in South Africa, as elsewhere, faced increased vulnerability due to the consequences of the COVID-19 response such as loss of income, challenges to access diagnostic testing, healthcare services and TB medication. To mitigate the socio-economic impact of the pandemic, especially among the most vulnerable, the South African government expanded social assistance programmes by creating the Social Relief of Distress grant (SRDG), the first grant for unemployed adults in South Africa. Our study investigated how TB patients experienced the COVID-19 pandemic and the ensuing socio-economic fallout, how this affected their health and that of their household, income and coping mechanisms, and access to social assistance. We interviewed 15 TB patients at a health facility in Cape Town and analysed data thematically. To situate our findings, we adapted the United Nations' conceptual framework on determinants of vulnerability and resilience during or following a shock such as climate shocks or pandemics. We found increased vulnerability among TB patients due to a high exposure and sensitivity to the COVID-19 shock but diminished coping capacity. The loss of income in many households resulted not only in increased food insecurity but also a decreased ability to support others. For the most vulnerable, the loss of social support meant resorting to begging and going hungry, severely affecting their ability to continue treatment. In addition, most participants in the study and especially the most vulnerable, fell through the cracks of the most extensive social assistance programme in Africa as few participants were accessing the special COVID-19 SRDG. Targeted social protection for TB patients with a heightened vulnerability and low coping capacity is urgently needed. TB patients with a heightened vulnerability and low coping capacity should be prioritized for urgent assistance.

publicly as study participants did not give consent to share the anonymised data publicly after the study. The original application to the ethics committee did not include an explicit request to participants to share data. Anonymised data is available after reasonable request to the SAMRC Ethics Committee for study replication. Please find the contact details for the SAMRC Ethics Office: Research Integrity Office Secretariat, Ms A Labuschagne (E-mail: Adri.Labuschagne@mrc.ac. za).

**Funding:** Funding for the study was provided by the Health Systems Research Unit at the South African Medical Research Council, as well as the Tampere TB Foundation to LV. The funding sources had no involvement in the design of the study; collection, analysis and interpretation of data; writing of the article; decision to submit it for publication.

**Competing interests:** The authors have declared that no competing interests exist.

## Introduction

While COVID-19 surpassed Tuberculosis (TB) as the world's leading infectious disease killer in 2020, TB remains the second biggest infectious disease killer, with more than 1.5 million people having died from the disease in 2020 [1]. The COVID-19 pandemic has had a severe impact on the fight against TB worldwide [1]. Health workers, testing machines, laboratories and health centers were diverted from existing diseases like TB to fight the new pandemic [2]. Reduced access to TB diagnosis and treatment in 2020 resulted in close to half of the people ill with TB not accessing care, and TB deaths, for the first time in over a decade, increased worldwide [1].

South Africa has been severely affected by the COVID-19 pandemic, but also by the subsequent government-imposed lockdowns. At the time of our study, February 2021, South Africa had counted more than 1,5 million cases of COVID-19 and close to 50,000 deaths [3]. In addition, approximately 257,543 excess deaths were reported in South Africa from early May 2020 to mid-September 2021, which may potentially be due to the indirect effects of COVID-19 [4]. The drop in economic activity due to the subsequent lockdowns lead to a record-breaking unemployment rate of 44.4% [5], the highest in the world, and the loss of employment and income led to two thirds of South African households experiencing food insecurity, while weekly child hunger increased from 8% in 2018 to 14% by March 2021 [6]. Women in South Africa were hit the hardest by the economic impact of COVID-19 with 30% of those employed having lost their job, compared to 22% of men [6].

People with TB in South Africa, as elsewhere, faced an increased vulnerability due to the consequences of COVID-19 and the ensuing government response. South Africa is one of the top eight countries with the highest burden of TB worldwide [1]. In 2019, an estimated 360 000 South Africans fell ill with TB, of which 22 000 patients died [7]. The COVID-19 pandemic affected TB services across South Africa with TB equipment, health personnel and resources being diverted to fight COVID-19, and healthcare facilities and laboratories severely overburdened, resulting in a 41% decrease in monthly active TB case notification for 2020 compared to 2019 [8]. Following the global trend, TB mortality in South Africa also increased during 2020 and is estimated to remain higher for at least the next 5 years [1].

In response to the COVID-19 crisis and regulations, most governments around the world expanded their social protection systems as a means of providing relief to vulnerable individuals and households [9]. South Africa's government expanded its non-contributory social assistance system (unconditional cash transfers or social grants) through a temporary increase in the amounts of all social grants, including the Disability Grant. In addition, a special COVID-19 Social Relief of Distress grant (SRDG) was introduced for unemployed working age adults who were destitute because of the virus [10]. The SRDG was the first grant in South Africa for people aged 18 to 59 years old who have no income and hence the first grant to start addressing the high levels of unemployment and poverty in the country [11]. The SRDG benefit of R350 ($23) per person per month provided a lifeline, however small, for many families [11]. Yet, while close to 6 million people were receiving the SRDG by December 2020, many challenges of exclusions and inequitable access were reported particularly for women, foreign nationals and people living in rural areas [12].

In addition to the SRDG, TB patients can apply for the Disability Grant (DG) provided to people with a physical or mental disability that are unfit to work for a minimum of six months [13]. Previous studies have however indicated that few TB patients access the DG [14–16].

Our study explored how TB patients experienced the COVID-19 pandemic and government-imposed response such as lockdowns, and how this affected their health and that of their household, income and coping mechanisms, and access to social assistance. To situate our

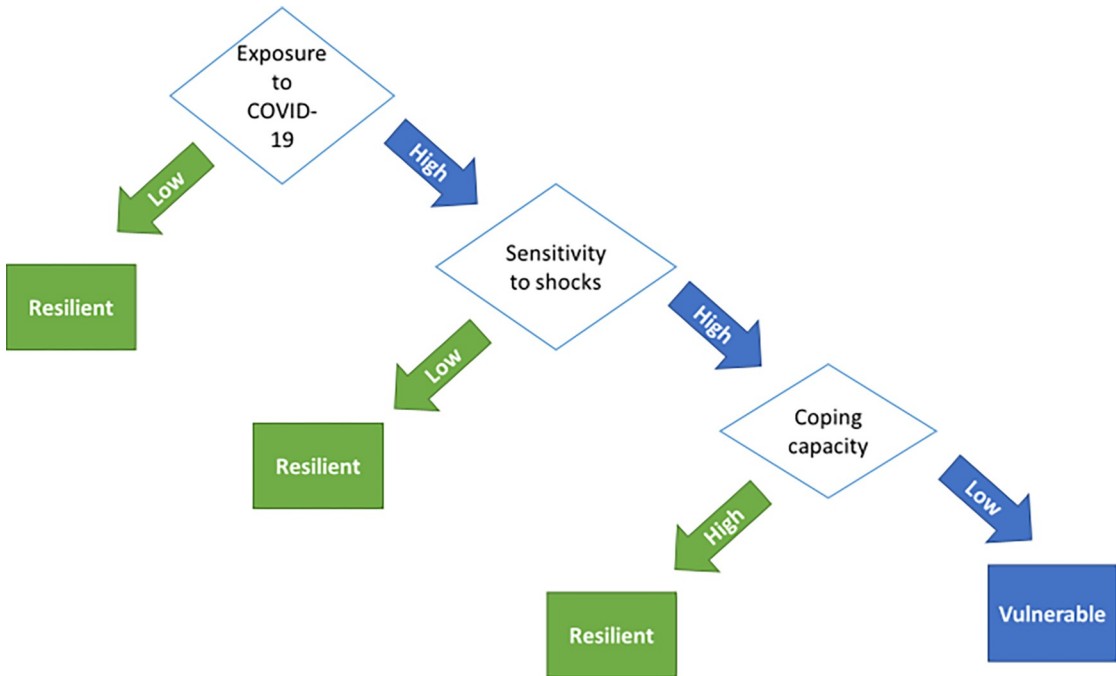

**Fig 1. Conceptual framework for determinants of vulnerability and resilience during or following a shock** [17].

findings, we have adapted the United Nations' conceptual framework on determinants of vulnerability and resilience during or following a shock such as climate shocks or pandemics. The framework highlights exposure to shocks, sensitivity to shocks and coping capacity as determinants of vulnerability and resilience (see Fig 1) [17]. While this framework has been primarily used to assess household vulnerability to (climate) shocks, we will use it in our discussion to frame the vulnerability of TB patients during COVID-19. The framework recognizes the potential of coping strategies and mechanisms such as social assistance and social support, to counter vulnerability and increase resilience during periods of shock.

## Methods

### Study design

We conducted an exploratory study using qualitative interviews to focus on the lived experiences and perceptions of people with TB during the COVID-19 pandemic in South Africa.

### Context and setting

Interviews with TB patients during the COVID-19 pandemic were conducted between February and March 2021 in a primary healthcare (PHC) clinic in a township on the outskirts of Cape Town, Western Cape, South Africa. The township has a high population density (16958 persons/km2), with more than 40% of inhabitants living in informal dwellings such as shacks, 22.2% of the working age population reporting no income, and 72% of households reporting an income of less than R3200 ($212) per month [18].

The Western Cape Province had a high burden of both TB as well as COVID-19 at the time of the study. The province had the second highest number of confirmed drug-sensitive TB (DS-TB) cases (44,816) and third highest number of confirmed drug-resistant TB (DR-TB) cases (1608) of the 9 provinces in South Africa in 2018 and 2017 respectively [19]. Within the

Western Cape province, the Cape Town Metropolitan Municipality which includes the study site, had the highest number of confirmed DS-TB cases (25,556) and second highest number of confirmed MDR-TB cases of all 52 districts in South Africa in 2018 and 2017 respectively [19]. At the time of the study, the Western Cape had just come out of a second COVID-19 wave and counted 281.223 cumulative cases by 20 March 2021, the third highest in the country [20].

In the Western Cape province, primary healthcare (PHC) clinics provide treatment and care for DS-TB and DR-TB free of charge. The Cape Town Metropolitan Municipality has more than 130 community-based clinics, ensuring healthcare services close to patients' homes. The study clinic has a dedicated "TB room" that diagnoses, initiates on treatment, and provides Directly Observed Therapy (DOT).

Pre-COVID-19, the South African government provided social assistance to those deemed vulnerable i.e. children, the elderly and the sick and disabled [13]. In response to the COVID-19 pandemic, the government of South Africa introduced a special COVID-19 Social Relief Distress grant (SRDG) of R350 ($23) in May 2020 [21]. The SRDG is the first grant for unemployed adults between the ages of 18 and 59 years old. See Table 1 for an overview of available grants in South Africa.

## Recruitment and sampling

A sample size of 15 interviews was chosen for the study. We followed Malterud's model to ascertain information power to guide the sample size [23]. According to the model, considerations about study aim, sample specificity, theoretical background, quality of dialogue, and strategy for analysis should determine whether sufficient information power will be obtained with less or more participants included in the sample [23]. As the aim of our study is narrow, the combination of participants is highly specific for the study aim, the study is supported by established theory, the interview dialogue is strong, and the analysis includes in-depth exploration of narratives, a smaller sample size is warranted. In addition, findings from a recent study showed that a sample size of 9–17 interviews is sufficient to reach saturation for relatively homogenous study populations [24], and previous research by the authors with the same study population reached saturation around 15 interviews [16]. A convenience sample of 15 TB

**Table 1. Social grants, number of beneficiaries and value of each grant at the time of the interviews, February–March 2021 [12, 13, 21, 22].**

| Grant | Eligible persons | # Beneficiaries | Value of grant |
|---|---|---|---|
| Child Support Grant (CSG) | a primary caregiver of a child under the age of 18 years old. | 7 262 081 (13 452 285 children) | R445 ($30) /month |
| Old Age Grant (OAG) | People 60 years and older. | 3 722 675 | R1 860 ($125) /month |
| Disability Grant (DG) | A person unable to provide for his or her maintenance due to a mental or physical disability. | 997 752 | R1 860 ($125) /month |
| Grant In Aid (GIA) | People receiving an Old Age, Disability or War Veteran's grant and unable to care for themselves who need full-time care from someone else. | 267 912 | R445 ($30) /month |
| Foster Care Grant (FCG) | Foster parent. | 230 785 | R1 040 ($70) /month |
| Care Dependency Grant (CDG) | A parent, primary caregiver or a foster parent of a child who requires and receives permanent care or support services due to his or her (physical or mental) disability. | 146 931 | R1 860 ($125) /month |
| War Veteran Grant (WVG) | War veterans | 40 | R1 880 ($126) /month |
| Social Relief of Distress Grant (SRDG) | Persons in such a dire material need that they are unable to meet their families' most basic needs. | 5 774 151 (of 9 955 717 applications processed) | R350 ($23) /month |

patients was drawn from TB patients attending a primary healthcare clinic that provides treatment and care for people with TB. TB patients, generally, come to the clinic early in the morning and wait to be seen by the nurse. Following the check-up by the nurse, they are asked to wait in the waiting area to see the doctor. We requested nurses to refer any patient older than 18 years and on treatment for at least two weeks to the research assistant in the adjoining room after their check-up. In a private room, the research assistant explained the study and read the informed consent forms together with the patient. If the patient agreed to participate in the study, the patient was asked to sign the consent form and a pre-interview participant profile form was completed with the patient after consent. The information collected included patients' socioeconomic characteristics, TB and/or COVID-19 history in the household, and receipt of social grants. After completing the participant profile form, the interview would commence.

## Data collection

We conducted semi-structured interviews with 15 TB patients. Interviews were conducted in isiXhosa in a private room in the clinic. Interviews explored how both TB and COVID-19 affected the patient's life and that of the household they lived in, their experience of accessing healthcare services, the use of coping mechanisms, and experience of applying for social grants such as the special COVID-19 Social Relief of Distress Grant (SRDG) and Disability Grant (DG). All interviews were recorded, transcribed verbatim and translated into English by a professional translation company and checked against the original recording to ensure accuracy by the bilingual researcher.

Following each day of interviews, LV and WZ discussed the interviews, noting initial thoughts and meanings, refining questions, and noting emerging areas for further research. Transcripts and notes were deidentified and stored on the South African Medical Research Council's (SAMRC) secured servers.

## Data analysis

Transcripts and notes were analysed using thematic analysis [25] to identify and interpret patterns and themes in the qualitative data. Transcripts were read and re-read to allow for familiarisation and to start the process of open coding. Coding was performed inductively on MS Excel. Quotes were interpreted and condensed for meaning, then organized into codes and themes (see Table 2). Preliminary analysis was performed by LV and reviewed by WZ and SA following which the analysis was revised.

## Ethical considerations

The study was approved by the Human Research Ethics Committee of the South African Medical Research Council (EC015-8-2017). All participants were given an information sheet and informed consent form which was read together with the participant and explained in detail. Participants were informed about the purpose of the study, procedures involved, risks and benefits of the study, and their rights as participants. The right to decline participation was emphasised, as well as an assurance given that the decision not to participate would not affect the services they receive at the clinic or any other government institution. Participants were given an assurance of confidentiality and strict protection of collected data. Following the detailed explanation, participants were asked to sign the consent form. All 15 participants in the study signed the consent form.

**Table 2. Example of coding.**

| Number, gender, DS/DR-TB | Quote | Condensed meaning | Code | Theme |
|---|---|---|---|---|
| 1M-DR | they say that it is R800 to test for Corona, it's a lot of money | was told R800 for COVID-19 test | didn't test for COVID-19, too expensive | COVID19 testing |
| 1M-DR | here there was never an offer to test you? | never tested for COVID-19 | didn't test for COVID-19, too expensive | |
| 1M-DR | I only tested for HIV, the results were negative | tested neg for HIV | HIV- | comorbidities |
| 1M-DR | Is there a person that you know in your community or family that was diagnosed with Corona and then passed away? No, there isn't any. No, no one died even with my cousins | no COVID-19 deaths in family | no COVID-19 deaths in family | disease/death in the family |
| 1M-DR | No, it was a lot for almost two months with a lot of back-to-back funerals | many COVID-19 deaths (funerals) in the area | COVID-19 deaths | |
| 2F-DS | I live with my grandmother, aunt and my cousins, my aunt's children. It is the seven of us. | household of 7: grandmother, aunt, 1 student, 4 children | large household | household |
| 2F-DS | We survive on my grandmother's social grant and aunt's, since she receives some for her children, she is not working. And there is also NSFAS | income: OAG, aunt grant, 4xCSG, 1x NSFAS | Household income from grants (CSG + NSFAS) | income |
| 2F-DS | I am studying At West Coast College | patient studies | - | employment |
| 2F-DS | I have four cousins. The first one is 15. And the other one is 12. The other one is 6. And the last one is 1 year. | 4 children under 18 | children in the household | household |

## Researcher characteristics and reflexivity

The study was conducted by three researchers, WZ, LV and SA. All have masters' qualifications in social sciences, with WZ and SA having obtained doctoral training in public health and social policy research. All are experienced in qualitative research. WZ is a black Xhosa-speaking female, LV is a white foreign English-speaking female living in South Africa and SA a white foreign English-speaking female living away from South Africa. LV conceptualized the study, WZ conducted the interviews, LV lead the analysis, and WZ and SA participated in the analysis.

In the process of conducting the interviews, WZ understood that she brought with her the insider-outsider perspective and experience to each interview and to the research process as a whole, as someone who both embodies the lived experiences of Black low-income South Africans, but who, through her educational qualifications and her current middle class positionality in the socio-economic strata, is removed from the immediate economic conditions and experiences of participants residing in a township. WZ used this awareness to forge a connection with respondents and ensured that it did not cloud the data collection process.

LV was aware that being white, middle-class and foreign, despite long residence in South Africa, might impede her understanding of participants' experiences and perceptions. Previous qualitative research with people with TB, however, familiarised her with issues and challenges for people with TB. In addition, WZ who conducted the interviews narrated each participant's story in detail during the feedback session after the interview and ensured to detail the participant's experiences but also her own observations of the emotional and mental undertones of each story as well as the physical appearance of the participant.

SA brought with her a background of living and working in South Africa for 14 years but having since moved away. Her work in South Africa focused on evaluating TB services. She has strong views on equity, social justice and believes in the potential of social protection systems to support health globally.

## Results

We interviewed 15 TB patients between the beginning of February and the end of March 2021 (see Table 3). During this period, South Africa had just come out of its second wave of COVID-19, and lockdown rules had been eased from alert level 3 to alert level 1. All participants, however, had started feeling ill and been diagnosed in the previous year, 2020.

The health impact of both TB and COVID-19 was severe on both patients and their households and families. Most TB patients had a history of TB and some had already had it twice before. Those with no prior history of TB were relatively young, between the ages of 20–30 years old, seemingly indicating that the lack of previous TB is likely because they are still young. Several participants also mentioned someone in the household or close family recently having been sick with TB or currently showing symptoms of TB. Despite the high rate of recurring TB and presence of TB among close contacts, contact tracing and screening was not performed and only a few participants were asked to bring family members, mostly children under the age of 5 years old, to the clinic for screening.

In addition to the health impact of TB on both patients and their households, during COVID-19, four participants had tested positive for COVID-19 but also mentioned someone else in the household having had COVID-19. Four participants also mentioned a recent death in the family without being specific about the cause of death.

Several themes emerged from the interviews relating to symptoms and illness before diagnosis, access to healthcare services, impact of COVID-19 on employment and income, impact of COVID-19 on the household, food insecurity, social support and coping mechanisms, and access to social assistance. In this article, we focus on the following themes relating to the personal impact of COVID-19 on TB patients and their household, and access to social assistance:

- "We struggle to buy bread"—Loss of income and food insecurity

- "She is also struggling"—loss of social support

- Limited social assistance for TB patients

- Falling through the cracks–no support for the most vulnerable

**Table 3. Details of participants.**

|  | Female (9) | Male (6) |
| --- | --- | --- |
| DS-TB | 8 | 4 |
| DR-TB | 1 | 2 |
| Had COVID-19 | 3 | 1 |
| TB patients that lost income due to COVID-19 and or TB (from formal or informal work) | 2 | 4 |
| Household members of TB patients that lost income due to COVID-19 (from formal or informal work) | 4 | 2 |
| Received SRDG | 1 | 1 |
| Received DG | 0 | 0 |
| Age range of participants |  |  |
| 20–30 years old | 5 | 1 |
| 31–40 years old | 3 | 0 |
| 40 years and older | 1 | 5 |
| History of previous TB | 6 | 5 |
| No history of TB | 3 | 1 |

### "We struggle to buy bread"—Loss of income and food insecurity

Most participants were unemployed, but some earned a little cash from informal work such as handyman, collecting recycling, guarding parked cars, braiding hair, and selling chicken feet and ginger beer. This income, however, was lost for those participants that became too ill to work.

In addition, those that hadn't already lost their income due to TB, now also lost their income due to the COVID-19 restrictions. One participant had lost his formal employment while five other patients reported to have lost income either due to the TB illness, COVID-19 restrictions or both. For example, one of the participants earned a little income braiding hair but found she was too tired to braid hair when she was sick with TB. When she started getting better, she struggled to get customers due to COVID-19.

> *They (customers) are not coming as before because people don't have money—a lot of people —due to this COVID, so people do their hair very seldom. (female, 32 years old)*

In addition to their own loss of income, six participants mentioned one or more members of their household losing income, severely impacting the household budget. With many participants relying on their household during their illness, the loss of income in the household affected them as well.

The loss of income in the household affected the most basic needs of the household such as food and electricity. While food insecurity was prevalent among TB patients before COVID-19, significantly more participants spoke of food insecurity plaguing the household every month during COVID-19. About half of participants testified that the household ran out of food before the end of the month resulting in meals being skipped to make the available food last.

> *So you would find out the food is not enough for all of us or ends before time. We would struggle to buy bread for lunch. (. . .) When we are approaching towards the end of the month we run out of it, but we know some day we will have food (. . ..) So if we ate breakfast in the morning we have to take three or two slices because we have to leave some slices for the kids in order to eat when they get hungry again and so on. After that you will get your next meal in the evening. (female, 36 years old)*

Several participants reported that the lack of food made it difficult to take treatment because of the side-effects of taking treatment on an empty stomach.

> *Sometimes I would feel like I wish to drink the pills, but there would be no food to eat. Then you end up not knowing what to do, I end up not drinking the pills because the pills are powerful for me to drink them on an empty stomach. They can even cause me to collapse. (female, 34 years old)*

Female participants testified that when there was food in the house, they would feed the children first, sometimes leaving nothing for them to take their medication with.

> *I would stay and not take the pills when there is no food because sometimes my friend would buy me porridge, but the kids eat it. (female, 32 years old)*

## "She is also struggling"—loss of social support

Most participants were supported by their household or family with care and especially food. Some participants had moved to a family member or friend that was caring for them while they were sick. Family members were also looking after children while the mother was in hospital or too sick to care for them. But households of TB patients, however, were struggling due to income loss caused by COVID-19 and therefore less able to support with care and especially food, as one patient attests:

*When she does have something, I eat what they eat. Okay, but not all the time because she also is struggling, because she has two kids too. (female, 32 years old)*

Several participants, especially those without family support, were forced to ask for help including borrowing money and asking for food from neighbours and friends. The level of asking or "begging" for help was widespread and frequent during COVID-19:

*It's hard, it's very, very, very difficult. I don't know where to start or where it's hard. It's hard to ask . . .. and begging and begging and begging and begging and begging and begging. (male, 40 years old)*

In the absence of support, one mother had to turn to her 9-year-old child to help her with care practices like bathing herself while acutely ill:

*I would ask the boy (9 years old) to place the bath vessel on the floor, tell him to be careful not to get burnt by the hot water, to pour the water in the vessel. It troubled me greatly. (female, 32 years old)*

## Limited social assistance for TB patients

At the time of the interviews, the COVID-19 Social Relief of Distress Grant (SRDG) had been provided to close to 5.8 million people for close to 10 months. As the name suggests, the SRDG was created to provide relief to those that were in dire need due to the socio-economic consequences of COVID-19. While the need for social assistance was high for most participants in our study, few of them received the SRDG. All but three participants had applied for the SRDG, but only two were successful in their application. Participants reported several exclusion criteria and challenges with the application process.

Female participants with children receiving the Child Support Grant (CSG) reported being rejected as receipt of another grant was an exclusion criteria for the SRDG. Caregivers of children receiving a CSG were given a CSG top-up of R300 ($20) per child in May 2020, followed by a R500 ($33) allowance per caregiver from June to October 2020. However, the Caregivers Allowance ended in October 2020 and caregivers were not allowed to apply for the SRDG, resulting in female participants with children being excluded from receiving the grant.

In addition to the automatic exclusions, challenges with several government databases also resulted in rejections. Three participants reported their application for the SRDG was rejected because their name appeared on the Unemployment Insurance Fund (UIF) database, while another participant was rejected because of an employee tax certificate (IRP5) on the South African Revenue Service (SARS) database. UIF unemployment benefits can be claimed when an employer terminates employment and the employee has been contributing to the UIF during their employment. The application must be made within six months of termination of

employment. Thereafter, while UIF benefits can no longer be claimed, the former employee's name remains in the database, leading to an automatic rejection of the SRDG application. Similarly, having an old IRP5 on file at SARS also resulted in an automatic rejection.

None of the 4 participants appealed the decision. Being sick and lacking resources and energy to fight the outcome of their application, they gave up.

*I have been trying to check for it, but now when I came back, I went to check for it again and they said that I am not even in the system. I got tired in the process. While focusing on one thing they tell you to start again, hooo, no! (male, 49 years old)*

Another participant's wife couldn't apply for the SRDG because her ID had been lost when their shack burned to the ground. The application for a new ID required a trip to the Home Affairs office (all of which were closed during COVID-19) and a fee of R140 ($8) which the couple couldn't afford.

Another participant was still waiting for an outcome six months after submitting the application.

In addition to the SRDG, TB patients can apply for a Disability Grant (DG), provided to people with a physical or mental disability that are unfit to work for a minimum of six months. The application requires a medical report and recommendation by a medical doctor appointed by the South African Social Security Agency (SASSA). While the majority of participants expressed the need for social assistance and seven participants had applied or were in the process of applying, none of the participants in the study was receiving the DG at the time of the interview. Eight participants reported not having applied for the DG, five of whom incorrectly assumed they weren't eligible while two others were unsure about how the process worked as the clinic did not display nor make any information available on the application process for a DG. Instead, several participants attested that one must ask the doctor or nurses for information which according to one participant was not an easy task:

*I do want it my sister. I do not want to lie but I am scared of asking them. And how it works. I have always wanted to ask about it, but I get scared of them. (male, 53 years old)*

The lack of easily available information about the DG resulted in many misconceptions about eligibility for the DG and the application process. One participant thought the DG was for people who had had TB at least twice. Another participant didn't think she could apply because she was not sick enough.

*Since I was never seen being seriously sick, when I came to the day hospital, I was not terribly sick. I was walking on my own so I thought I was not going to get it (the disability grant), since they can see that I still have some energy as well. (female, 21 years old)*

For those that persevered and continued with the application process, a long and cumbersome process awaited which was only made more frustrating and difficult because of COVID-19. SASSA offices were unable to handle the volume of applications due to backlogs caused by the closure or disruptions in services and the need to ensure compliance with COVID-19 guidelines such as social distancing, resulting in endless queues and applicants being turned away. One participant testified how she was turned away at the SASSA office three times:

*I wanted to apply for it, but at SASSA things are hard due to COVID. They tell us to wait, wait, wait and promise to call, but they never call. The social worker said I should go to (the*

*SASSA office in) town. When I got to town it was full and only a few people were taken inside. When I went there again, they said that it is closed at SASSA due to COVID. (female, 32 years old)*

SASSA's inability to continue services during COVID-19 forced applicants to take drastic actions. One participant spent the night outside the SASSA office, with many others, to be in the front of the queue the next day. After a long and cold night, huddled together with others to stay warm, she was turned away because the office had reached its maximum capacity allowed during COVID-19. The following week the participant spent another long cold night in front of the SASSA office and was turned away again because by now the application letter had expired. On her third attempt and with a new application letter, she made it into the SASSA office and could finally submit her application for the DG. After one of the long cold nights outside in front of the SASSA office, however, she tested positive for COVID-19. At the time of the interview, she had not yet received the result of the application.

The increased difficulty to apply for the DG during COVID-19 also resulted in increased costs for applicants at a time when resources were already stretched, making it increasingly difficult for applicants to continue the process. The repeated return trips to the SASSA office for the TB patient and often someone to accompany them as they are weak from the TB illness, incurred increased transport and food costs. One of the participants attests she had to borrow money to finance the repeated trips:

*While there, you would then be turned back. We spend money to go there. I was very ill when I went there. I was not strong yet. I was trying to borrow from people. I was fed up already. We use money to travel all the way. At SASSA when they speak there that "You should come back tomorrow" when they speak as if we always have the money and one spends the whole day there and just when the day is almost finished they tell you late in the day to come back tomorrow, when you are already very hungry. (female, 32 years old)*

Participants reported the cost of transport for one trip to be between R40 ($2,5) to R60 ($4) and anything between 1 to 5 trips would be needed to finalise the application, totaling anywhere between R40 ($2,5) to R300 ($20) for a $23 value SRDG or $125 value DG. Combined with the loss of time and health risks, many gave up. Some, however, driven by sheer desperation continued:

*"The problem is that I'm desperate for money, I need the money."*

### Falling through the cracks–no government support for the most vulnerable

The struggles of TB patients to access social support grants seemed to mount particularly on the most vulnerable. The story of a young pregnant woman highlights how it is possible to fall through the cracks of an extensive social protection system and encounter extreme distress. The young pregnant woman was unemployed and without family to support her as she had migrated from the Eastern Cape to Cape Town in search for work. Her job appointment at the municipality fell through at the start of the pandemic and her boyfriend and father of the unborn child had also been unable to find employment during COVID-19. They both had to rely on the boyfriend's sister, who herself had to survive on a Disability Grant, and a neighbor to help with food. Despite the help, she regularly went without food and recalls how she had gone for two whole days without food while being pregnant. She spoke of the traumatizing experience of feeling and knowing that the unborn baby in her belly was as hungry as she was.

At the time of the interview, the participant was mere days away from giving birth but still weighted less than 50kgs.

> *Yho it is not easy my sister. Because you often think about the unemployment we are facing, what are we going to do with this (unborn) child? What is he/she going to eat? As I do not have food—as in now—and then there is him/her in my tummy, she/he is hungry right now inside my tummy, so it is traumatising.*
>
> *(female, 34 years old)*

Despite the existence of several support mechanisms that could have supported her, she accessed none. 1) She received instant porridge at the clinic, which is provided to patients with a BMI of less than 18.5, three months after starting her TB treatment. After receiving it for two months, the porridge was out of stock at the clinic and she stopped receiving it. This means that although she was eligible for 6 months of food support, because she was identified late as being underweight and due to the frequent stock-outs of porridge at the clinic, she only received 2 months of food support. 2) She applied for the SRDG but because she was receiving a child support grant (CSG) for her oldest child, she was automatically rejected for the SRDG. The CSG for her oldest child was sent directly to the grandmother in the Eastern Cape who was caring for the child while the patient had migrated to Cape Town in search for work. 3) The patient was also eligible for a disability grant (DG), yet none of the nurses nor doctor at the clinic thought to inform her about, nor approve her for this form of social assistance. The doctor at the clinic who holds considerable discretion over the decision to provide a DG did not use his discretion to recommend a pregnant female, weighing less than 50kg, with TB for social assistance.

## Discussion

The biomedical interactions and co-infections between TB and COVID-19 [26–29], as well as the impact of the COVID-19 pandemic on the TB epidemic in terms of incidence, TB diagnosis and enrolment on treatment, and disruption of health services for TB patients has been widely described, discussed and modelled [8, 30–34]. Our study reports on the socio-economic impact of COVID-19 on TB patients, their coping mechanisms such as social support and their challenges to access government-provided social assistance during the COVID-19 pandemic in Cape Town, South Africa. Our study found increased vulnerability among TB patients due to increased loss of income caused by COVID-19 imposed lockdowns, and increased loss of social support as households, families and neighbours were also affected by COVID-19, yet limited access to social assistance in the form of the special COVID-19 Social Relief of Distress Grant (SRDG) or Disability Grant (DG).

The United Nations' framework for determinants of vulnerability and resilience during or following a shock, highlights exposure to shocks, sensitivity to shocks and coping capacity as determinants of vulnerability and resilience (see Fig 1) [17]. Using this framework to guide analysis of the findings of our study, we found that during the COVID-19 pandemic TB patients faced an increased vulnerability due to high exposure and sensitivity to the COVID-19 pandemic, especially for those where coping capacity was also affected by COVID-19 (see Fig 2). Exposure to the COVID-19 shock was high for TB patients, as for the majority of the South African population, as the virus had a high prevalence in South Africa but also the government-imposed lockdowns were reported as the most severe in the world [35]. In addition, patients in our study showed a high sensitivity to the COVID-19 shock as most had no stable

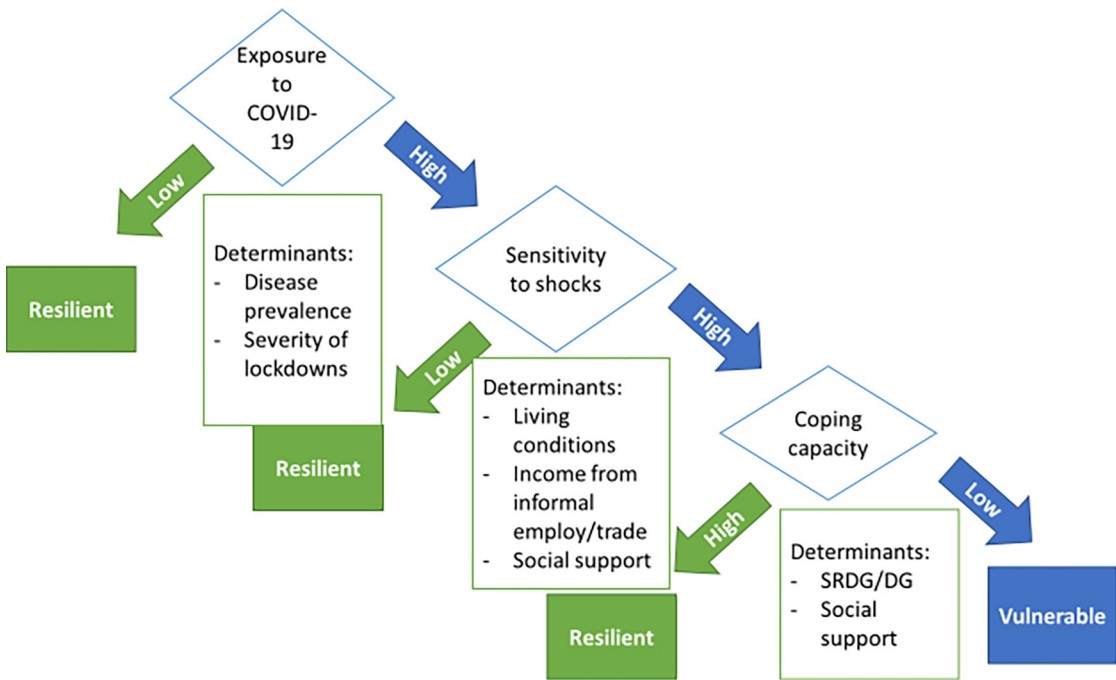

**Fig 2. Determinants of vulnerability and resilience experienced by TB patients in the study.**

income and were faced with poor living conditions such as crowded, informal living spaces and food insecurity.

TB is known to disproportionally affect the most economically disadvantaged in any society [36] but also worsens poverty through medical and non-medical costs (direct costs), and loss of income by reducing patients' physical strength and ability to work (indirect costs) [14, 36, 37]. For many participants in our study the loss of income due to TB was exacerbated by the loss of income in the household due to the COVID-19 pandemic which led to an increase in food insecurity and a decrease in the ability of many households to support TB patients. This corresponds with findings from the few available studies on the financial impact of COVID-19 on TB patients. A study in India found that the number of TB patients' households with zero household income increased from 6% before lockdown to 54% during the strict lockdown [38]. The study further found indications that the income loss for households and poor restoration of income after lockdown will have had a serious impact on the nutrition of the TB patients and families. A study in Cairo, similarly, found that the early COVID-19 lockdown was associated with a rise in catastrophic costs for TB patients [39].

Food insecurity has been one of the major consequences of the COVID-19 pandemic and consequent lockdowns in South Africa. The Coronavirus Rapid Mobile Survey (NIDS-CRAM) that tracked key outcomes during the COVID-19 pandemic, reported that 37–47% of the population indicated that their households ran out of money to buy food and 16–22% indicated that their households experienced hunger over the May/June 2020 to February/March 2021 period. TB patients rely heavily on their households and families for care and most often food [16]. With household budgets decimated due to the economic fallout of COVID-19, resulting in increased food insecurity in many households, care and food support for TB patients has been severely affected as well.

Social support, such as support by households, families, and neighbours, can increase coping capacity and counter vulnerability. Social support has also been shown to have a strong

influence on treatment adherence and even treatment outcomes [6–10]. The absence of social support, however, can have a negative impact on adherence [40–42], health-related quality of life [43] and even mortality [40–42, 44]. Our study found that families and households are the main source of care and support for many patients with TB, but also that they 'shield' patients with TB from hunger and probably worse treatment outcomes. The loss of income and increased food insecurity in many households caused by COVID-19, however, resulted in less social support being available for many TB patients. Those with little social support struggled tremendously with food insecurity and experienced hunger on a regular basis. The loss of income, combined with a loss of social support structures, became tragic for some. The loss of social support has been reported as one of the major consequences of COVID-19, and particularly its effect on mental health and isolation [45–48]. The effect of COVID-19 on social support for TB patients and the resulting loss of material support, however, has to our knowledge not been studied. Those without social support should be prioritized for support in the form of cash and in-kind transfers including food vouchers, parcels, and community feeding schemes.

Social assistance is another coping mechanism that can counter the increased vulnerability and build resilience. Our study, however, found that most TB patients were falling through the cracks of the most extensive social assistance programme in Africa and not accessing the special COVID-19 Social Relief of Distress Grant nor the Disability Grant. While receipt of the COVID-19 SRDG has reportedly been pro-poor with most of the individuals who received the SRDG being in the poorest quintiles [10], only two participants in our study were receiving the SRDG. Similar to issues reported by the Black Sash report, the most common reason for a declined grant application was conflict with different national databases (UIF, SARS or NSFAS). The faulty and outdated systems that SASSA used to make determinations and reject grant applications ended up disadvantaging many eligible applicants [49]. In addition, women with children did not access the SRDG, despite carrying the load of care and support for households, children, and those sick with TB.

Application for the DG has previously been shown to be a long, cumbersome and costly process [16, 50, 51]. During COVID-19 the already existing challenges were exacerbated, resulting in applicants incurring extra costs: 1) financial costs to cover transport for the multiple trips to government departments, 2) the cost of time lost having to queue repeated times to access services, and 3) the cost to the person's health when being forced to spend the night in front of the SASSA office. For TB patients, especially those that are severely ill and weakened by the disease, these costs can become insurmountable, restricting access to much needed social assistance.

For many of our participants, high exposure to COVID-19, combined with a high sensitivity to shocks and little to no coping capacity resulted in increased vulnerability. Targeted social protection to the most vulnerable populations such as TB patients is urgently needed. TB patients with a heightened vulnerability and low coping capacity should be prioritized for urgent assistance.

## Strengths and limitations

Our study was explorative and conducted with TB patients attending a clinic in Cape Town. The results can therefore not be generalized to all TB patients in South Africa or elsewhere. Our findings, however, highlight issues likely to be relevant to many TB patients, especially those living in similar socio-economic circumstances and affected by the COVID-19 pandemic. In addition, our study has captured multiple challenges experienced by TB patients in accessing government provided social assistance that could help inform government's social

protection strategy as well as the current discussion on the introduction of the Universal Basic Income Grant in South Africa.

## Conclusion

Our qualitative study investigated how TB patients experienced the COVID-19 pandemic and the ensuing socio-economic fallout, how this affected their health and that of their household, income and coping mechanisms, and access to social assistance. To situate our findings, we adapted the United Nations' conceptual framework on determinants of vulnerability and resilience during or following a shock and found increased vulnerability among TB patients due to a high exposure and sensitivity to the COVID-19 shock but diminished coping capacity. The loss of income in many households resulted not only in increased food insecurity but also a decreased ability to support others. For the most vulnerable, the loss of social support meant resorting to begging and going hungry, severely affecting their ability to continue treatment. In addition, most participants in the study and especially the most vulnerable fell through the cracks of the most extensive social assistance programme in Africa as few participants were accessing the special COVID-19 Social Relief of Distress Grant and none accessed the Disability Grant, mainly due to issues with government administration systems. Targeted social protection to the most vulnerable populations such as TB patients is urgently needed. TB patients with a heightened vulnerability and low coping capacity should be prioritized for urgent assistance.

## Acknowledgments

The authors thank the staff of the healthcare facility where the interviews were conducted for their collaboration and efforts to accommodate the research team. Last but not least, we thank the research participants for their participation and openness.

## Author Contributions

**Conceptualization:** Lieve Vanleeuw, Wanga Zembe-Mkabile, Salla Atkins.

**Formal analysis:** Lieve Vanleeuw, Wanga Zembe-Mkabile.

**Funding acquisition:** Lieve Vanleeuw, Wanga Zembe-Mkabile, Salla Atkins.

**Investigation:** Lieve Vanleeuw, Wanga Zembe-Mkabile.

**Methodology:** Lieve Vanleeuw, Wanga Zembe-Mkabile, Salla Atkins.

**Project administration:** Lieve Vanleeuw.

**Supervision:** Wanga Zembe-Mkabile, Salla Atkins.

**Writing – original draft:** Lieve Vanleeuw.

**Writing – review & editing:** Lieve Vanleeuw, Wanga Zembe-Mkabile, Salla Atkins.

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
