## [Decision Letter · Decision Letter 0]

20 Apr 2022

PGPH-D-22-00517

Falling through the cracks: increased vulnerability and limited social assistance for TB patients and their households during COVID-19 in Cape Town, South Africa.

Dear Dr. Lieve Vanleeuw,

Thank you for submitting your manuscript to PLOS Global Public Health. After careful consideration, we feel that it has merit but does not fully meet PLOS Global Public Health’s publication criteria as it currently stands. Therefore, we invite you to submit a revised version of the manuscript that addresses the points raised during the review process.

Reviewers' comments:

Reviewer's Responses to Questions

**Comments to the Author**

1. Does this manuscript meet PLOS Global Public Health’s publication criteria? Is the manuscript technically sound, and do the data support the conclusions? The manuscript must describe methodologically and ethically rigorous research with conclusions that are appropriately drawn based on the data presented.

Reviewer #1: Yes

Reviewer #2: Partly

2. Has the statistical analysis been performed appropriately and rigorously?

Reviewer #1: N/A

Reviewer #2: N/A

3. Have the authors made all data underlying the findings in their manuscript fully available (please refer to the Data Availability Statement at the start of the manuscript PDF file)?

Reviewer #1: Yes

Reviewer #2: Yes

4. Is the manuscript presented in an intelligible fashion and written in standard English?

Reviewer #1: Yes

Reviewer #2: No

5. Review Comments to the Author

Reviewer #1: Assessment

□ Summary statement:

South Africa is one of the most unequal societies in the world, with high levels of unemployment and poverty. South Africa remains the epicentre of the HIV pandemic. HIV positivity and overcrowding contribute to high prevalence of TB in the local population, with majority of affected being the poor people of indigenous African descent. Due to multiple reasons, the rates of poor compliance to treatment resulting in drug resistant TB is also alarmingly high. COVID-19 pandemic with the hard lockdowns, made the matters significantly worse and exposed the most vulnerable, poor, unemployed people to further poverty, hunger, malnutrition, and unintended noncompliance to treatment. It is a painful and sad reality that, despite of having the best social support network in the entire African continent, many fell through the cracks and did not receive intended and expected social relief. This situation was even worse for people suffering from TB. It is recognised that management of chronic infectious diseases like HIV and TB suffered seriously during the COVID-19 pandemic, as majority of the resources were diverted towards diagnosis and management of COVID-19.

This is a cross-sectional phenomenological exploratory study using qualitative interviews conducted in a clinic in the Western Cape province of South Africa during the height of the COVID-19 pandemic, to investigate how TB patients experienced the pandemic and the ensuing socio-economic fallout, how this affected their health and that of their household, income and coping mechanisms, and access to social assistance. They interviewed 15 TB patients at a health facility in Cape Town and analysed data thematically. They have scientifically demonstrated that targeted social protection to the most vulnerable populations such as TB patients is urgently needed in South Africa and TB patients with a heightened vulnerability and low coping capacity should be prioritized for urgent assistance.

□ Topical relevance and novelty of the work:

This manuscript is relevant to the journal to which it is submitted.

The topic is relevant to the general understanding of the effects of the current socioeconomic situation on the treatment of one of the most prevalent infectious diseases, TB and suggests a solution to rectify it.

□ Background

It sets the stage for understanding the origins and importance of the work.

□ Aims

The primary aim and secondary aims are clearly stated.

□ Methods

The cross-sectional phenomenological exploratory study method is appropriate to address the stated aims.

□ Statistics

As it a qualitative study, there is no need to apply the statistical methods.

□ Results

The results are presented in a logical way, and they address the stated aims.

□ Discussion

The discussion addresses the results in a systematic way, pointing out the strengths and weaknesses of the study. I would have liked to know, how many of these 15 patients were HIV positive and how many were on antiretroviral (ARV) therapy. This data is not available.

□ Limitations

The limitations of the study, its design, its methods and reported data are clearly indicated by the authors. There are no ethical concerns.

□ Figures and Tables

The included figures and tables are appropriate for the manuscript. The tables are easy to read and understand.

The legends for the figures and tables are appropriate and easy to read.

□ References

There are appropriate number of timely references that support the background and discussion.

□ Title/Keywords

These are appropriate for the manuscript.

□ Abstract

The abstract captures the most salient points of the study including key results and conclusions.

□ Readability

The manuscript is easy to read, and the use of language is appropriate.

Following are the specific minor areas where I offer corrections for grammar, spelling, or punctuation corrections:

Page 23 Line 351 Use word “people” instead of short form used “ppl”

Page 28 Line 475 Use upper case F for the first word in the sentence “for”

□ Approval

I have noted that appropriate research approval was obtained.

□ Conflicts

There are no conflicts of interest, and it is clearly stated.

Reviewer #2: SUMMARY

An interesting article highlighting the plight of Tuberculosis (TB) patients in a public clinic in the context Covid 19 pandemic using the cross-sectional phenomenological exploratory design. The qualitative study used semi-structured interviews to focus on the lived experiences and perceptions of patients suffering from Tuberculosis during the Covid 19 pandemic in Cape Town, South Africa. The results reveal how the ripple effects of the pandemic affected the TB patients’ health and that of their household by pointing out how the loss of income/ inability to get income due to hard lockdown resulted to food insecurity which eventually led to defaulting of treatment and/or some family member exhibiting TB/ Covid 19 symptoms because there was little/no contact tracing done. Additionally, patients had to cope with their food insecurities through begging for food, skipping meals to stretch the available scarce food resource to last longer. The article also demonstrated how most of the patients who direly needed the government social assistance (Disability Grant and Social Relief of Distress Grant) fell through the cracks as they and their care givers were let down by ignorance, fear, cost of application, and faulty and outdated national data systems that led to rejection of eligible applicants.

MAJORS ISSUES

Introduction

There are sections with statements that lack referencing.

Line 52 and 53 says the patients were turned away but it doesn’t indicate who turned them away

What do other studies say about loss of income and support systems, and health impact to TB patients and their households in the context of Covid 19 if any?

Methodology

While the sample size is logical, it is not clear as to how the researchers got to the sample of 15 participants. Did they reach saturation by the 15th patient or they just took a logical number?

Most TB diagnosis, treatment and treatment review in the public health centres are managed by the nurses and not all TB patients get referred to the doctors. What was the inclusion and exclusion criteria for the participants?

In the data collection, the researchers noticed and considered new areas for further interviews. Did they follow this up with the participants?

It is not clear how the researchers ensured accuracy of the translation without any Bias as there was only one bilingual researcher for the research context who had the sole responsibility of translation.

Results

Line 242,243 &244. The researchers indicate that about half of the participants were admitted to hospital for long Periods for severe illness and lost out on income as a result. This is conflicted by table 3 as it shows that only one patient lost income due to TB illness.

Line 247 indicates that several participants working informally lost income due to lockdown restrictions while table 3 shows only 2 participants.

MINOR ISSUES

The article needs editing.

Acknowledge the sources of the data e.g. table 1

For the headings in table 1. Please put a dash if there is no information to fill in or a zero if appropriate.

Consent forms. It’s the participants who are supposed to give informed consent after explanation by signing the consent form.

Line 109 Covid 19 pandemic not “epidemic.”

6. PLOS authors have the option to publish the peer review history of their article (what does this mean?). If published, this will include your full peer review and any attached files.

**Do you want your identity to be public for this peer review?** For information about this choice, including consent withdrawal, please see our Privacy Policy.

Reviewer #1: **Yes: **DR PRATIMA CHITNIS

Reviewer #2: No

We look forward to receiving your revised manuscript.

Kind regards,

Muhammed Olanrewaju Afolabi, MD, MPH, PhD

Academic Editor

Journal Requirements:

1. Your co-author, Wanga Zembe-Mkabile -wanga.zembe@mrc.ac.za, has not confirmed authorship of the manuscript. We have resent them the authorship confirmation email; however please check that the above email address for them is correct and follow up personally to ensure they confirm. 

Please note that we cannot proceed your manuscript  until we have received confirmations from all co-author. 

2. Please note that your Data Availability Statement is currently missing the repository name and direct link to access each database. If your manuscript is accepted for publication, you will be asked to provide these details on a very short timeline. We therefore suggest that you provide this information now, though we will not hold up the peer review process if you are unable.

---

## [Decision Letter · Decision Letter 1]

5 Jul 2022

Falling through the cracks: increased vulnerability and limited social assistance for TB patients and their households during COVID-19 in Cape Town, South Africa.

PGPH-D-22-00517R1

Dear Ms Lieve Vanleeuw,

We are pleased to inform you that your manuscript 'Falling through the cracks: increased vulnerability and limited social assistance for TB patients and their households during COVID-19 in Cape Town, South Africa.' has been provisionally accepted for publication in PLOS Global Public Health.

Best regards,

Muhammed Olanrewaju Afolabi, MD, MPH, PhD

Academic Editor

Reviewer Comments (if any, and for reference):

Reviewer's Responses to Questions

**Comments to the Author**

1. If the authors have adequately addressed your comments raised in a previous round of review and you feel that this manuscript is now acceptable for publication, you may indicate that here to bypass the “Comments to the Author” section, enter your conflict of interest statement in the “Confidential to Editor” section, and submit your "Accept" recommendation.

Reviewer #1: (No Response)

Reviewer #2: All comments have been addressed

2. Does this manuscript meet PLOS Global Public Health’s publication criteria? Is the manuscript technically sound, and do the data support the conclusions? The manuscript must describe methodologically and ethically rigorous research with conclusions that are appropriately drawn based on the data presented.

Reviewer #1: Yes

Reviewer #2: Yes

3. Has the statistical analysis been performed appropriately and rigorously?

Reviewer #1: N/A

Reviewer #2: N/A

4. Have the authors made all data underlying the findings in their manuscript fully available (please refer to the Data Availability Statement at the start of the manuscript PDF file)?

Reviewer #1: Yes

Reviewer #2: Yes

5. Is the manuscript presented in an intelligible fashion and written in standard English?

Reviewer #1: Yes

Reviewer #2: No

6. Review Comments to the Author

Reviewer #1: If authors have not collected data on how many patients were HIV positive, I would like it to be mentioned in the limitations of the study.

Reviewer #2: The article has a few grammatical errors (see highlighted areas) that render the sentences incoherent.

7. PLOS authors have the option to publish the peer review history of their article (what does this mean?). If published, this will include your full peer review and any attached files.

**Do you want your identity to be public for this peer review?** For information about this choice, including consent withdrawal, please see our Privacy Policy.

Reviewer #1: No

Reviewer #2: **Yes: **Onyango Peggy
